# Problem-Solving Treatment for People Recently Diagnosed with Visual Impairment: Pilot Randomised Controlled Trial

**DOI:** 10.3390/jpm12091431

**Published:** 2022-08-31

**Authors:** Afsane Riazi, Trefor Aspden, Gary Rubin, Gareth Ambler, Fatima Jichi, Laurence Mynors-Wallice, Miriam O’Driscoll, Kate Walters

**Affiliations:** 1Department of Psychology, Richmond American University London, London W4 5AN, UK; 2Department of Psychology, Royal Holloway, University of London, Egham TW20 0EX, UK; 3Institute of Ophthalmology, University College London, London EC1V 9EL, UK; 4Department of Statistical Science, University of London, London WC1E 6BT, UK; 5Biostatistics Group, University College London Hospitals/University of London Research Support Centre, University College London, London WC1E 6BT, UK; 6Dorset HealthCare University NHS Foundation, Poole BH17 0RB, UK; 7Department of Primary Care and Population Health, University College London, London NW3 2PF, UK

**Keywords:** problem-solving treatment, visual impairment, psychological well-being, quality of life, vision loss

## Abstract

Background: Problem-Solving Treatment (PST) has been used to treat and prevent depression in a variety of settings. However, the impact of PST on improving psychological well-being in those with recent vision loss remains unknown. The aim of this study was to evaluate whether PST may lead to better psychological well-being in people with recent vision loss through a pilot parallel-group randomised controlled trial. Methods: Participants who were diagnosed with visual impairment during the previous 3 months were randomly allocated to either an 8-week PST or treatment as usual (N = 61). Outcome measures were administered at baseline, 3, 6, and 9-months. Results: A linear mixed model demonstrated that PST significantly improved psychological well-being (measured by the Warwick Edinburgh Mental Well-being Scale) (treatment effect = 2.44; 95% CI = 0.40–4.47; *p* = 0.019). Significant improvements in the PST group for symptoms of distress, quality of life and self-efficacy were also observed. There was no significant difference in mobility. The treatment effect was consistent at all follow-ups. Attrition rate was low (13%). Conclusions: PST was associated with a significant and sustained improvement in a range of outcomes in people with recent vision loss. Further large scale RCT is now required.

## 1. Introduction

Epidemiological data in the UK indicates incidence of depression amongst older adults with visual impairment to be 13.5%, compared to 4.6% among older adults with normal vision [1]. However, major depression is not an expected outcome of vision loss, and many with low vision do not become clinically depressed [2,3]. Instead, adults with vision loss may experience distress around perceived control, fear of dependency, and perceived loss of ability to maintain meaningful personal connections and social roles [4]. Newly diagnosed people may receive mobility training or provision of devices designed to aid activities of daily living such as low vision aids, but these services do not routinely target psychological needs per se, and the impact of these services on quality of life are modest [5].

Problem-solving treatment (PST) is an established, brief, structured psychological intervention that specifically addresses individuals’ confidence, motivation and psychological well-being by teaching a rational and systematic approach to problem solving and addressing negative perceptions that may interfere with finding solutions [6]. It is an intervention in which individuals learn to use their own skills and resources to cope with both present and future problems [7]. PST has been used successfully in people with anxiety and depression within primary care [6,8]. Interventions based on problem-solving skills training has been demonstrated in various samples (e.g., [9,10,11,12,13]), including those with macular degeneration [14]. PST has potential as an effective tool in coping with the diagnosis of vision loss as focusing on the present may create situations that appear more manageable. PST-treated individuals with macular degeneration were able to develop compensatory strategies to continue pursuing valued activities, resulting in reduced levels of depression [14]. The concept of well-being depicts feelings of happiness and a sense of purpose which can remain even in the presence of mental illness, distress, or suffering [15,16,17,18]. The current study aimed to assess PST through a pilot RCT of PST plus treatment as usual (TAU) versus TAU for individuals newly diagnosed with visual impairment. We expected better well-being scores in participants undergoing PST plus TAU compared to those participants with TAU only.

## 2. Materials and Methods

### 2.1. Design

A single-blind, multicentre, parallel-group pilot RCT of PST plus usual care (PST group) versus Treatment As Usual (TAU group) for people newly diagnosed with visual impairment. This included an assessment of adherence, recruitment and retention.

#### Participants

Participants were eligible for inclusion if they were:

adults (≥18 years of age);community-dwelling (i.e., not living in a care home);diagnosed with severe, irreversible sight loss, or registered as sight impaired (partially sighted) or severely sight impaired (blind) within the previous 3 months.

Participants were excluded if they (i) participated in psychiatric or psychological assessment or intervention within the previous 3 months; (ii) had severe cognitive impairment (screened by the Six Item Cognitive Impairment Test [19]; whereby a score of ≥10 will result in exclusion); (iii) were severely hearing impaired (to a level that makes participation impractical); (iv) and/or had insufficient proficiency in English to impact on participation.

### 2.2. Procedure

Potential participants were identified through either community recruitment via five Country Council and London Borough Sensory Needs Services or through six NHS secondary services (Eye Hospitals/Eye Units) across London, South East and Central England. For the community recruitment, information about the study were sent via the appropriate Sensory Needs Service team approximately a week after the participants received their Certificate of Visual Impairment (CVI), along with a reply slip and pre-paid envelope. For the NHS recruitment, participants were provided the same information pack by an NHS-based Vision Support Officer (VSO), an ophthalmologist, or a research nurse. If they chose to participate, participants were screened for suitability by the telephone, and if deemed suitable, then a home visit was arranged for the Study Coordinator to obtain informed consent and the full assessment was conducted. This research adhered to the tenets of the Declaration of Helsinki.

### 2.3. Outcomes and Assessments

All assessments were administered by the Study Coordinator at week 1 (baseline), 3, 6 and 9 months. However, assessments at 3 and 6 months were limited to outcomes of well-being, symptoms of distress, mobility, and quality of life to reduce participant burden. The exceptions to this were in those instances where the 9 months follow-up were not possible, in which case the full assessments were conducted at 6 months instead. Baseline assessments were conducted at participants’ homes, while subsequent assessments were conducted over the telephone where possible.

The primary outcome was measured by the Warwick-Edinburgh Mental Well-being Scale (WEMWBS) [16]. This assesses positive mental health (mental well-being), including 14 positively worded items with five response categories. It covers most aspects of positive mental health (positive thoughts and feelings) currently in the literature, including both hedonic and eudaimonic perspectives. It has good interrater and test–retest reliability and is sensitive to change. Analysis was conducted once WEMWBS was transformed to interval-level measurement using Rasch analysis.

Secondary outcomes were as follows:

(i)psychological distress, as measured by the Hospital Anxiety and Depression Scale [20];(ii)functional mobility, measured by the Self-assessed Instrument for Perceived Visual Ability for Independent Mobility [21] and Life Spaces Questionnaire [22];(iii)quality of life, measured by the Impact of Vision Impairment Questionnaire [23] and the VISQOL [24];(iv)problem-solving ability, measured by the Social Problem Solving Inventory—Revised: Short [25];(v)self-efficacy, measured by the Generalized Self-efficacy Scale [26].

The reliability and validity of the above scales are reported in their respective publications. Self-reported information on health resource use over the previous 4 weeks were also gathered by a health resource use questionnaire developed for a previous study [8].

Total scores were computed using the recommended algorithm for each scale, with mean imputation used as appropriate for each scale scores, but not the total scores.

Our sample size of 120 patients was chosen to detect an improvement on the Warwick-Edinburgh Mental Well-being Scale of 5 points (assuming a SD of 8.8 points, a conservative estimate derived from a Scottish population survey [16]) with a two-sided 5% significance level and 80% power, given an anticipated dropout rate of 15%.

#### 2.3.1. Randomisation Procedures

Participants were randomly allocated with a 1:1 allocation to either PST or TAU, using stratified randomisation to balance severely sight impaired (blind) vs. sight impaired (partially sighted) across the groups. An independent web-based randomisation service was used which was administered by the study therapist, and the outcome assessor and study team was masked to group allocation. Blocking was used within strata with random block sizes of 2, 4 and 6. After randomisation the study therapist informed the participants of the group allocation. Participants could not be masked to group allocation because the intervention was psychosocial. However, they were reminded not to disclose which arm of the study they were assigned to at the beginning of each assessment, in an attempt to minimise detection bias.

#### 2.3.2. Intervention and Control Groups

In the PST group the participants took part in six PST sessions of 45–60 min over an eight week period. Sessions were delivered by a study therapist trained in PST by Dr. Mynors-Wallis, the developer of PST. The first 4 sessions took place weekly, and the last 2 sessions every two weeks. All PST sessions took place at the participant’s home. PST consisted of seven stages, with core elements of the structure present in all sessions. These stages represent discrete steps in either the treatment process (i.e., explanation of the treatment and its rationale, and evaluation of progress) or in the problem-solving process itself (i.e., identifying and defining problems, establishing achievable goals, generating solutions, evaluating and choosing the solution and applying the chosen solution) [27]. Typically in the first session the therapist gave the following simple explanation of the rationale of problem solving: emotional symptoms are caused by problems in living; if the problems are dealt with effectively the symptoms will improve; problems can be dealt with effectively by the technique of problem solving [28]. After this explanation the participant’s problems were identified and listed, and by discussion with the therapist they chose one problem as a focus of treatment for the first session [28]. The stages of problem solving referred above were explained to the participant by reference to the chosen problem, and in subsequent sessions further problems were dealt with in the same way [28]. Three months after completing the final PST session, the study therapist conducted a follow-up telephone booster session to review knowledge and skills of problem-solving. Participants in the PST group continued to receive routine medical care (such as further eye examinations) and rehabilitation (such as mobility training) depending on individual needs. In the TAU group participants also continued to receive routine medical care.

### 2.4. Statistical Methods

All analyses were carried out using all available data with participants in the groups to which they were originally randomised, and no imputation was performed for missing data. Baseline participant characteristics were described using means (SDs) or medians (interquartile range) for continuous measures, and proportions for categorical measures.

For the primary analysis, a linear mixed model was used to evaluate the effect of treatment on the primary outcome measure, the Rasch-transformed WEMWBS score at 3,6, and 9 months with fixed effects for treatment, time, severity of vision loss (stratification factor), and baseline Rasch-transformed WEMWBS score, and a random effect for patient. As a sensitivity analysis, the primary analysis was repeated using the raw WEMWBS scores as the outcome. Additionally, the primary analysis was repeated after including covariates for age, gender, and cause of visual impairment as fixed effects in the model, to adjust for observed imbalances in the baseline characteristics.

As a secondary analysis, we explored whether the effect of treatment changes over time by including an interaction between treatment and time as a fixed effect in the primary analysis model. We also explored whether the effect of treatment differed for the severity of vision loss groups in a similar manner. These secondary analyses should be only considered as exploratory as the study was not powered for them.

Linear mixed models were also used to evaluate the effect of treatment on the different secondary outcomes, with fixed effects for treatment, time, severity of vision loss, and baseline score, and a random effect for patient. All analyses were carried out using Stata 13.1.

Ethical approval was obtained through the NHS Research Ethics Committee (reference number 13/LO/0416). The trial registration number was ISRCTN87854656, and the UK Clinical Research Network Portfolio number was 14289.

## 3. Results

### 3.1. Recruitment and Retention

Figure 1 shows the CONSORT flow diagram of participants through the trial. In total, 954 people were contacted via the sensory needs teams and NHS recruitment sites. The overall response rate was 10.0% (Response rate in Sensory Needs Teams = 7.5%; NHS sites = 22.9%). 95 people with visual impairment were screened for eligibility. Of these, 61 participants were randomised into the PST or usual care groups of the POSITIVE Trial. Over the course of the study a total of 8 participants were lost to follow-up (withdrew from the study). This corresponds to an attrition rate of 13%, though this is a conservative estimate and the attrition rate was actually lower because some of these participants sill contributed information at the 3 month point.

### 3.2. Participants

Table 1 shows the characteristics of the sample. There were 26 males and 35 females aged between 29 and 98 years. The PST group had a slightly higher proportion of females (62.1%) than the no treatment group (53.1%) and was slightly younger on average (mean (SD): 70.9 (19.5) vs. 75.6 (15.8)) with lower (raw) WEMWBS scores (50.5 (7.2) vs. 45.7 (11.2)). The PST group also had higher HADS total scores (10.5 (6.9) vs. 13.5 (7.3)). A sensitivity analysis was therefore performed for the primary analysis to adjust for the imbalance in gender and age. Balance seems to have been achieved on other characteristics.

### 3.3. Primary Outcome Analyses

The estimated mean difference in Rasch-transformed WEMWBS score between the two groups (PST—treatment as usual) was 2.61 (95% CI: 0.27 to 4.95), 1.78 (−0.60 to 4.16), and 3.11 (0.53 to 5.69) at 3, 6, and 9 months, respectively, after adjustment for baseline score and severity of visual impairment (Table 2).

Overall, the PST group had an average Rasch-transformed WEMWBS score that was 2.44 (0.40 to 4.47, *p* = 0.019) points higher than that of treatment as usual group, after adjustment for baseline score and severity of visual impairment. Adding age, gender and cause of visual impairment to the analysis model resulted in the same conclusion (difference = 2.28: 95% CI: 0.25 to 4.32, *p* = 0.028) as did use of the raw WEMWBS score as the outcome (difference = 4.44: 95% CI: 0.05 to 8.93, *p* = 0.053).

There was no evidence to suggest that the effect of treatment changed over time (*p* = 0.50) (Figure 2).

### 3.4. Secondary Outcome Analyses

There was some evidence that HADS scores were lower for subjects receiving PST treatment with an estimated effect of 2.76 points (95% CI: 0.60 to 4.92, *p* = 0.012) for HADS Total. Additionally, there was evidence that IVIQ scores were higher for people receiving PST with an estimated effect of 0.23 units (raw score rather than Rasch score) (95% CI: 0.08 to 0.37, *p* = 0.0020). People receiving PST also had higher Self Efficacy scores with an estimated effect of 0.27 points (95% CI: 0.06 to 0.49, *p* = 0.011). There was little evidence of a difference for the remaining secondary outcomes (Table 3).

The descriptive statistics (means/sds) of the health resources questions for each group suggest that participants in the PST group reported fewer visits to the GP at Surgery (0.5 (0.6) vs. 0.8 (0.8)) and Nurse at Surgery (0.1 (0.3) vs. 0.4 (0.9)) at 6 months, as well as reporting fewer days in which they were unable to follow usual activities at 9-months (4.4 (8.9) vs. 13.7 (13.1)) (Appendix A).

## 4. Discussion

This trial demonstrated that PST is feasible for people newly diagnosed with visual impairment and that PST is potentially efficacious with regard to psychological well-being. Adjusting for baseline score and severity of visual impairment, the PST group had significantly better psychological well-being (our primary outcome), than the treatment as usual group. The PST group also demonstrated significantly better quality of life scores and depression and anxiety scores compared to the treatment as usual group. Therefore, the present research suggests that PST is an intervention that shows promise in improving wellbeing and quality of life, and reducing depression and anxiety, in people diagnosed as blind or partially sighted. Improvements in wellbeing, depression, and quality of life are maintained at the 9-month follow-up point, suggesting potential long-term effects of the intervention. Furthermore our secondary analyses suggested that PST is effective in both people who are partially sighted and those with severe visual impairment. There was no measurable impact of PST on mobility measured by two separate scales. This suggests that the effects of PST are mainly psychological as expected, and confidence in dealing with various problems do not readily translate to their levels of mobility. One explanation for this is that participants could explore any problems in their lives through PST, and so confidence in a particular aspect of their lives (e.g., relationships) may not easily translate to more confidence in their mobility levels.

In terms of acceptability and feasibility, although recruitment was slower than expected, the recruitment was better via NHS compared to community methods. Additional means of improving recruitment needs to be considered in any future trials [29]. In addition, in the community group, the study information and invitation letter were included with a pack of other information and documents sent routinely following a new registration of partial or severe visual impairment and it is quite likely that they were overwhelmed with information at this early stage of diagnosis. Retention was high once participants were recruited into the trial. This suggests that future trials need to employ more direct engagement with potential participants.

There were several limitations to this study. First, there was a difference in some of the baseline measures between the two groups. However, the sensitivity analyses demonstrate that this imbalance was not problematic. In our study we had no attention control group, and it is possible that the effects we saw were due to the extra attention from up to six home visits by the therapist, rather than the PST itself.

The sample size of our study, which although was sufficient to detect a significant difference in the two groups on several key variables, was still not large enough to allow a clear interpretation of the secondary analyses as well as the subgroup analyses. Thus it is not known whether self-efficacy is a mediator of change for the primary outcome (i.e., psychological well-being), although self-efficacy was significantly better in the PST group than the treatment as usual group.

Our study expands on the study by Rovner [14] by demonstrating the impact of PST on psychological well-being in people with visual impairment, and including participants from all adult age groups with vision loss from all causes. In particular, we have demonstrated that the booster sessions prolong the beneficial effects of PST, which in their study had diminished by 6-month follow up. Our findings can also be compared to other studies that have examined the impact of psychological interventions in this population. One study compared two interventions for depression, PST and referral to the patient’s physician, with a waitlist control group in people with sight loss and depressive symptoms [30]. This study found that depressive symptoms improved the most in the PST group and least in the control group, however the change was small and the uncertainty of the measurements relatively large [30]. In another study, a stepped care approach consisting of watchful waiting, guided self-help based on cognitive therapy, problem-solving treatment and referral to a general practitioner significantly reduced the risk of a depressive dysthymic and/or anxiety disorders at 24 months in visually impaired older adults [31]. These studies suggest that psychological interventions may improve mental health in people with low vision, but differences in methodology mean that conclusions cannot be easily drawn and further research is necessary.

This study was powered to detect a meaningful change in the primary outcome, that is, it was a fully powered pilot study. However, we did not recruit as many participants as intended, and therefore the results may need to be interpreted with some caution. A larger RCT is now required to enable us to better explore the secondary outcomes, subgroups and mediators. It would also provide better precision when it comes to estimating treatment effects. In our study the control group only received usual care. In future studies, active control groups should also be included. The data from this trial suggest that direct engagement with participants should be used to increase recruitment.

## Figures and Tables

**Figure 1 jpm-12-01431-f001:**
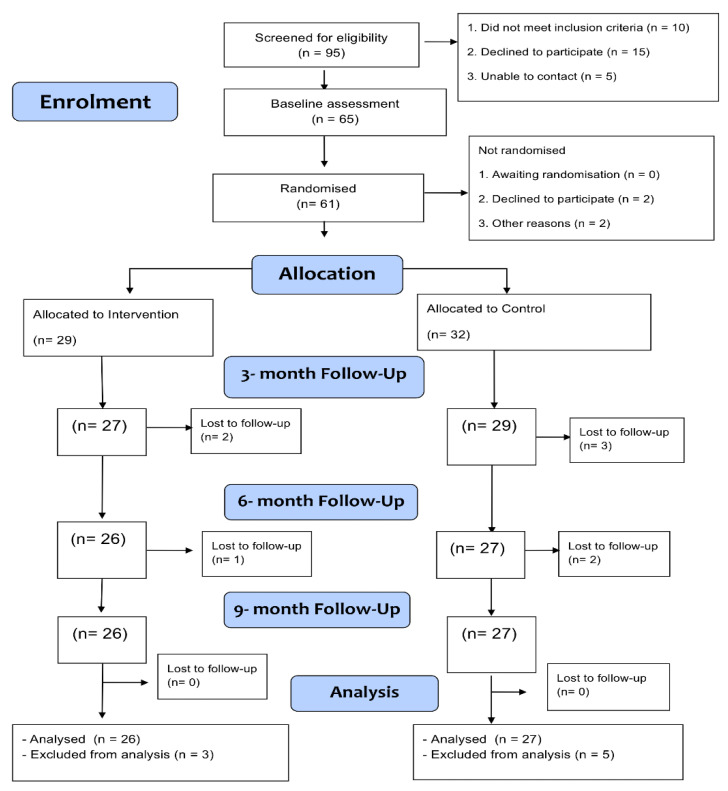
Consort Diagram.

**Figure 2 jpm-12-01431-f002:**
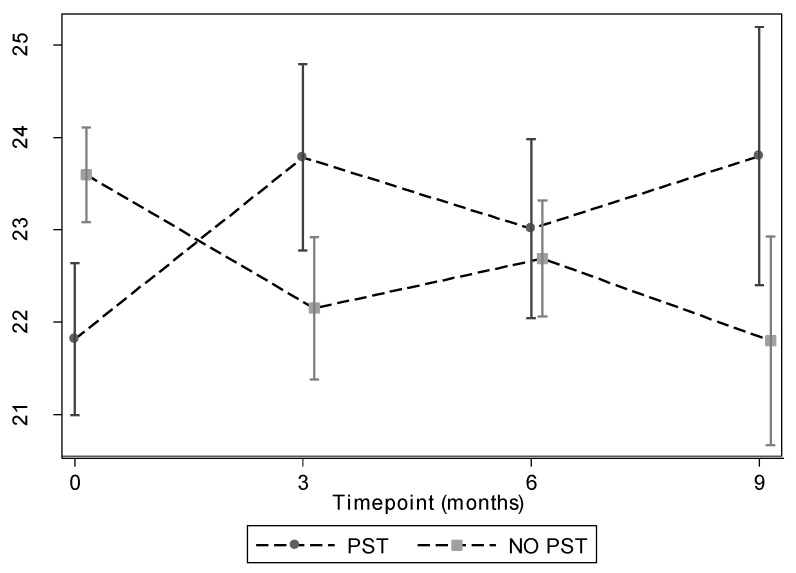
Mean Rasch WEMBS scores at each time point with error bars representing standard errors. Graphs broken down by treatment group.

**Table 1 jpm-12-01431-t001:** Baseline characteristics by treatment group. Continuous variables are summarized as mean (SD) and categorical variables are summarised as n (%).

	Problem-Solving Treatment (N = 29)	No Treatment (N = 32)
Age (years)	70.9 (19.5)	75.6 (15.8)
Female	18 (62.1)	17 (53.1)
Severely sight impaired (blind)	12 (41.4)	14 (43.8)
Cause of vision impairment: macular degeneration		
	13 (44.8)	15 (46.9)
WEMBS		
Raw score	45.7 (11.2)	50.5 (7.2)
Rasch score	21.8 (4.4)	23.6 (2.9)
HADS		
Anxiety	6.1 (4.0)	4.6 (4.1)
Depression	7.4 (4.4)	5.8 (3.2)
Total	13.5 (7.3)	10.5 (6.9)
IVIQ		
Mobility and independence	1.6 (0.5)	1.8 (0.5)
Emotion and well-being	1.9 (0.5)	2.1 (0.6)
Reading and information	1.7 (0.6)	1.9 (0.6)
Total score	1.7 (0.5)	1.9 (0.5)
Rasch total score	51.5 (6.5)	53.5 (6.1)
VISQOL		
Injure	0.7 (0.3)	0.7 (0.3) *
Cope	0.7 (0.3)	0.8 (0.2)
Friendships	0.9 (0.2)	0.9 (0.2)
Assistance	0.9 (0.2)	1.0 (0.1)
Roles	0.6 (0.4)	0.7 (0.4)
Confidence	0.8 (0.3)	0.8 (0.3)
Total Dimension Score	0.6 (0.3)	0.7 (0.2)
Life Spaces Questionnaire	4.9 (1.8)	4.7 (1.8) *
Independent Mobility		
Total score	2.7 (0.8)	2.4 (0.9)
Rasch score	−0.2 (1.0)	−0.6 (1.4)
SPSIRS total score	12.2 (0.9) n = 24	12.1 (0.5) n = 21
Self-efficacy score	2.9 (0.7)	2.9 (0.4)

* 1 missing item.

**Table 2 jpm-12-01431-t002:** Table showing mean (SD) Rasch-transformed WEMWBS scores at each time-point broken down by treatment group, with the overall treatment effect. The estimated treatment effects are adjusted for baseline score and severity of visual impairment.

WEMBS Score	PST	No PST	Treatment Effect (95% CI)	Overall Treatment Effect (95% CI)	*p*-Value
Baseline	45.7 (11.2)	50.5 (7.2)	-		
3 months	50.3 (9.6)	48.0 (9.8)	2.61 (0.27, 4.95)	2.44	
6 months	49.3 (11.2)	49.8 (6.9)	1.78 (−0.60, 4.16)	(0.40	0.019
9 months	50.2 (12.0)	47.0 (12.2)	3.11 (0.53, 5.69)	4.47)	

**Table 3 jpm-12-01431-t003:** Table showing the mean secondary outcome scores at each time-point by treatment group, with an overall estimate of treatment effect. This estimate is adjusted for baseline score and severity of visual impairment.

	PST	No PST	Treatment Effect (95% CI)	*p*-Value
HADS Anxiety				
At baseline	6.1 (4.0)	4.6 (4.1)	−1.29 (−2.39, −0.18)	0.022
At 3 months	5.1 (3.6)	5.3 (4.2)		
At 6 months	5.2 (3.9)	4.3 (3.1)		
At 9 months	4.7 (3.6)	5.6 (4.5)		
HADS Depression				
At baseline	7.4 (4.4)	5.8 (3.2)	−1.33 (−2.62, −0.05)	0.042
At 3 months	5.5 (3.0)	6.4 (3.6)		
At 6 months	6.7 (4.4)	6.2 (2.0)		
At 9 months	5.9 (3.7)	6.5 (3.5)		
HADS Total				
At baseline	13.5 (7.3)	10.4 (6.9)	−2.76 (−4.92, −0.60)	0.012
At 3 months	10.6 (5.6)	11.7 (6.9)		
At 6 months	11.9 (7.2)	10.4 (4.1)		
At 9 months	10.7 (6.2)	12.1 (7.4)		
IVIQ Total Score				
At baseline	1.7 (0.5)	1.9 (0.5)	0.23 (0.08, 0.37)	0.002
At 3 months	1.8 (0.5)	1.8 (0.4)		
At 6 months	1.9 (0.5)	1.8 (0.4)		
At 9 months	1.9 (0.5)	1.8 (0.5)		
IVIQ Rasch Total Score				
At baseline	51. 5 (6.5)	53.5 (6.1)	3.50 (1.49, 5.51)	<0.001
At 3 months	53.0 (7.0)	51.8 (5.7)		
At 6 months	54.2 (7.9)	51.9 (4.6)		
At 9 months	54.1 (7.4)	52.5 (6.0)		
VISQOL Total Dimension Score				
At baseline	0.6 (0.3)	0.7 (0.2)	0.07 (−0.02, 0.16)	0.112
At 3 months	0.6 (0.3)	0.6 (0.3)		
At 6 months	0.7 (0.2)	0.7 (0.3)		
At 9 months	0.6 (0.3)	0.6 (0.3)		
Life Spaces Questionnaire Score				
At baseline	4.9 (1.8)	4.7 (1.8)	0.22 (−0.31, 0.75)	0.418
At 3 months	4.8 (1.8)	4.7 (1.5)		
At 6 months	4.9 (2.0)	4.3 (2.0)		
At 9 months	4.6 (2.2)	4.4 (2.0)		
Independent Mobility Total Score				
At baseline	2.6 (0.8)	2.4 (0.9)	−0.05 (−0.29, 0.18)	0.650
At 3 months	2.6 (0.8)	2.3 (0.8)		
At 6 months	2.6 (0.8)	2.4 (0.6)		
At 9 months	2.7 (0.9)	2.5 (1.0)		
Independent Mobility Rasch Score				
At baseline	−0.2 (1.0)	−0.6 (1.4)	−0.11 (−0.46, 0.25)	0.550
At 3 months	−0.3 (1.2)	−0.7 (1.3)		
At 6 months	−0.3 (1.2)	−0.5 (0.9)		
At 9 months	−0.1 (1.2)	−0.3 (1.3)		
SPSIRS Total Score				
At baseline	12.2 (0.9)	12.1 (0.5)	0.14 (−0.09, 0.38)	0.239
At 6 months	3.8 (0.7)	3.6 (0.7)		
At 9 months	3.7 (0.9)	3.3 (0.8)		
Self Efficacy Score				
At baseline	2.8 (0.7)	2.9 (0.4)	0.27 (0.06, 0.49)	0.011
At 6 months	3.1 (0.7)	3.0 (0.5)		
At 9 months	3.1 (0.8)	2.7 (0.5)		

## Data Availability

Request for data can be directed to afsane.riazi@richmond.ac.uk.

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
