# Peer review of "Problem-Solving Treatment for People Recently Diagnosed with Visual Impairment: Pilot Randomised Controlled Trial"

_jpm, 2022, doi:10.3390/jpm12091431_

Round 1

Reviewer 1 Report

Thank you for the opportunity to review this work. This is a high-quality study that has the potential to impact common practice and improve wellbeing outcomes for those newly diagnosed with visual impairment.

One part that needs some improvement is the Discussion as it makes little reference to previous research targeting psychological outcomes among those suffering by visual impairment. Problem-solving treatment is a well-established intervention so it will be interesting to see whether other psychological interventions, if any, have been tested with this kind of patient group and how their findings compare with the findings of this study.

Furthermore, in lines 95-110 where the authors explain the scales that they use I would recommend a presentation per scale as that would improve the readability of those paragraphs.

Reviewer 2 Report

Thank you for the opportunity to read your paper. I have made some comments below:

Abstract 

Please include your outcome measures and data analysis. 

Background- the introduction is too short. Can you describe the concept of psychological well-being briefly? Can you describe how it is a PST intervention? 

Line 54- “The current study aimed to assess PST through a pilot RCT of PST plus treatment as usual (TAU) 55 versus TAU for individuals newly diagnosed with visual impairment”. Can you please describe your objective regarding the outcomes of the PST intervention? 

2. Materials and Methods- Did you follow a guideline for RCT? Consort? 

Outcome measures, please inform validity and reliability studies for all measures.  

PST sessions- Could you provide a table with the description of all sessions, including the objectives for each one? 

Line 223- correct punctuation. 

The discussion has only one reference. It would be good to discuss with the literature (I understand you don’t have much about PST); however, you can bring some evidence about visual impairment and self-efficacy, psychological well-being and so on.  
